# DiffusedSplitFed: Latent Diffusion and global feature fusion meet Split Federated medical image segmentation

## Abstract

Federated Learning (FL), Split Learning (SL), and Split Federated Learning (SplitFed) are emerging paradigms in privacy-preserving medical image analysis. FL enables multiple clients in collaborative model training without raw data exchange, while SL reduces client-side burden by partitioning the model between client and server. SplitFed combines the strengths of both but often faces limited representation power and semantic loss at the client-server interface, affecting both performance and privacy. The intermediate features and gradients transmitted can still reveal patterns from the original data, making them vulnerable to reconstruction attacks. This poses serious privacy risks, especially in sensitive domains like healthcare. This study proposes **DiffusedSplitFed**, the first SplitFed framework integrating Latent Denoising Diffusion Models (LDDMs) at both forward and backward split points to obfuscate transmitted representations. We design and compare three architectural variants (V1-V3) that explore dual conditioning and global feature fusion on segmentation performance, privacy preservation, and deployment complexity. We evaluated our framework on multiple medical imaging datasets, demonstrating significant segmentation performance while ensuring privacy and robustness compared to traditional SplitFed, state-of-the-art generative baselines, and privacy resilience baselines. We also provide a theoretical convergence guarantee. Our results underscore the potential of latent diffusion and global fusion for privacy-aware, high-fidelity medical image analysis. The implementation is available at: `https://anonymous.4open.science/r/DiffusedSplitFed`.

## 1 Introduction

Deep learning has advanced the field of medical imaging, enabling more precise identification of anatomical structures and pathological regions across various image modalities Shen et al. (2017). However, high-performance medical imaging models require large-scale annotated datasets that are often distributed across multiple medical institutions. The legal, ethical, and privacy-related regulations make it difficult to centralize data for collaborative model training. Federated Learning (FL) McMahan et al. (2017), Split Learning (SL) Gupta & Raskar (2018), and their amalgamation—Split Federated Learning (SplitFed) Thapa et al. (2022)—offer decentralized training solutions that allow institutions to collaboratively train models without sharing confidential raw data. However, conventional SplitFed still suffers from several critical limitations that impact both performance and privacy. The transmission of intermediate features and gradients can leak semantic information, enabling attacks such as reconstruction, model inversion, or gradient leakage to recover sensitive data or labels. These risks are especially critical in medical imaging, where data confidentiality is paramount.

The existing privacy-preserving architectures in SplitFed predominantly focus on statistical techniques such as differential privacy Wei et al. (2020), and cryptographic techniques such as homomorphic encryption Yang et al. (2023); Liang et al. (2025), secure multi-party computation Thapa et al. (2021), adversarial feature transformation/obfuscation Jiang et al. (2024), and other privacy-enhancing methods Thapa et al. (2021); Hukkeri et al. (2025). Although these approaches address privacy loopholes and provide sufficient theoretical guarantees Wei et al. (2020); Yan et al. (2024), they often introduce significant computational overhead. Furthermore, they may fail to preserve semantic fidelity in high-dimensional tasks such as medical image

segmentation. As an innovative solution, we propose utilizing the diffusion models for privacy preservation in SplitFed training.

Originally designed for image generation and segmentation, diffusion models use Denoising Diffusion Probabilistic Models (DDPMs) Ho et al. (2020) to learn complex data distributions by progressively reversing noise through a stochastic denoising process Ho et al. (2020). FedDDPM Peng et al. (2025) introduced an FL framework for DDPMs, where clients collaboratively train a diffusion model focusing on image generation. FedLMG Yang et al. (2025) proposed a one-shot FL framework that uses local models to guide conditional diffusion sampling for new clients with no labelled data. FedDEO Yang et al. (2024), FedDM Vora et al. (2024), and FedBip Chen et al. (2025) leveraged client-specific or communication-efficient generative models. Phoenix Stanley Jothiraj & Mashhadi (2024) trained an unconditional generative diffusion model in a distributed manner. CollaFuse Allmendinger et al. (2024) and FedDiffuse de Goede et al. (2024) introduced SplitFed diffusion frameworks. In medical segmentation, DDPM- and LDDM Rombach et al. (2022)-based approaches such as MedSegDiff Wu et al. (2024), SegDiff Amit et al. (2021), BerDiff Chen et al. (2023), GMS Huo et al. (2024), LSegDiff Vu Quoc et al. (2023), and LDSeg Zaman et al. (2024) have demonstrated high-fidelity, anatomically consistent mask generation. Concurrently, privacy-aware diffusion frameworks Tun et al. (2023) and privacy attack studies using DDPMs Gu et al. (2024) emphasize the relevance of noise-aware models in secure federated training.

In SplitFed settings, diffusion models offer an additional privacy advantage by actively injecting noise in split points. This deliberate noise injection helps to obfuscate sensitive information, making it difficult for adversaries to recover original inputs from transmitted features. Meanwhile, the learned denoising process retains sufficient semantic richness for downstream tasks. This dual capability makes diffusion models particularly well-suited for SplitFed learning. However, the use of latent diffusion models within SplitFed networks for privacy-preserving medical image segmentation remains largely unexplored. Bridging this gap is crucial for the deployment of scalable and practical deep learning systems in real-world clinical applications.

In this paper, we introduce DiffusedSplitFed, a novel Split Federated Learning framework that integrates LDDMs with dual conditioning and global feature fusion to enable privacy-preserving and context-aware medical image segmentation. In summary, our key contributions are fourfold:

- This is the first study to incorporate LDDMs in SplitFed with dual conditioning and global feature fusion, applied to both forward and backward passes to obfuscate latent features for privacy preservation.

- We propose and analyze three architectural variants for DiffusedSplitFed: V1 (dual conditioning with global fusion), V2 (global fusion only), and V3 (lightweight latent diffusion), to study the trade-offs between privacy, utility, and complexity.

- We provide a comprehensive privacy–utility trade-off analysis, supported by quantitative privacy metrics, segmentation performance evaluations, and a theoretical convergence guarantee for the proposed DiffusedSplitFed framework.

- We extensively validate the proposed variants on HAM10K Tschandl et al. (2018); Tschandl (2018), Blastocysts Lockhart et al. (2019), and FHPsAOPMSB (Fetal ultrasound) Lu et al. (2022); Kuş & Aydin (2024) datasets, showing that DiffusedSplitFed outperforms Baseline SplitFed and state-of-the-art (SoTA) generative methods in segmentation performance and privacy resilience.

Section 2 introduces the methodology, Section 3 presents experiments, Section 4 presents comparisons with SoTA methods, Section 5 outlines the theoretical convergence analysis, and Section 6 concludes the paper.

## 2 DiffusedSplitFed methodology

### 2.1 Enhancement over Baseline SplitFed

Our SplitFed architecture, based on Thapa et al. (2022) (Fig.1), partitions a UNet into front-end (FE), server-side (SS), and back-end (BE) components. Clients receive copies of the global FE and BE models

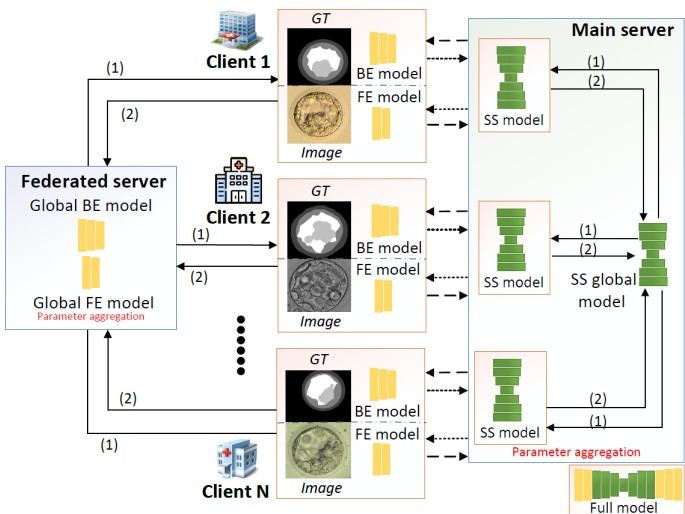

Figure 1: SplitFed uses multiple clients, each training a split UNet. (1) The federated server sends global FE and BE models, and the main server sends the SS model to clients. (2) Clients return trained FE and BE models to the federated server, and SS model copies to the main server. Long-dashed arrows show feature flow; short-dashed arrows indicate gradient flow. GT denotes ground truth.

from the federated server and the SS model from the main server. Each client performs local training over several epochs, after which models are aggregated using federated averaging McMahan et al. (2017) and redistributed for the next round of local training. This process is repeated for multiple communication rounds until convergence. To address the key limitations of this baseline architecture—such as information loss at split points and vulnerability to reconstruction attacks—we modified the SplitFed architecture, integrating latent diffusion mechanisms and global feature fusion to improve both privacy preservation and semantic consistency during clients' local training.

## 2.2 Architectural variants of DiffusedSplitFed

DiffusedSplitFed comprises three architectural variants (V1-V3), which differ in the way features and gradients are exchanged across the split points. The forward pass architectures of V1–V3 are shown in Fig. 2(a)–(c), while their backward passes are illustrated in Fig. 2(d) for V1 and in Fig. 3 (Appendix A) for V2 and V3. The corresponding forward pass training procedures are detailed in Algorithms 1, 2, and 3, respectively.

### 2.2.1 DiffusedSplitFed-V1: Latent diffusion with dual conditioning and global feature fusion

DiffusedSplitFed-V1 employs conditional latent diffusion, leveraging ViT Dosovitskiy et al. (2020)-based noisy label features during the forward pass and noisy image features during the backward pass to enhance semantic context and robustness. During the forward pass, the ViT encoder ($\text{ViT}_y$) is located on the CS-BE, where it receives the input labels ($y$) and extracts the corresponding ViT features ($\text{ViT}_y(y)$). Forward diffusion is applied to $\text{ViT}_y(y)$, forming noisy ViT features ($z_{\text{vit\_noisy}}$). $z_{\text{vit\_noisy}}$ are then combined with the clean features from the CS-FE ($z_{\text{FE}}$), forming the combined features ($z_{\text{combined\_FE}}$). $z_{\text{combined\_FE}}$ are then denoised using $\text{LDDM}_1$, generating local synthetic features of the FE ($LF_{\text{syn\_FE}}$). The reconstruction loss at $\text{LDDM}_1$ ($l_{\text{recon1}}$) is recorded. $LF_{\text{syn\_FE}}$ are then combined with the global synthetic features ($GF_{\text{syn\_FE}}$), which are the FE features of the most recent communication round. The combined feature set $z_{\text{OUT\_FE}}$ is sent to the SS model to carry on the forward pass. The noisy ViT features are combined with the clean features from the SS ($z_{\text{SS}}$). The combined feature set ($z_{\text{combined\_SS}}$) is denoised using the $\text{LDDM}_2$, generating local synthetic features of the server ($LF_{\text{syn\_SS}}$). The reconstruction loss at $\text{LDDM}_2$ ($l_{\text{recon2}}$) is recorded. $LF_{\text{syn\_SS}}$ are then combined with the global synthetic features ($GF_{\text{syn\_SS}}$), which are the SS features of the most recent communication round. The combined feature set ($z_{\text{OUT\_SS}}$) is then sent to the BE model to complete the forward pass. The final predictions ($\hat{y}$) are obtained, segmentation loss is recorded ($l_{seg}(\hat{y}, y)$), and

optimization continues, minimizing the combined loss function of the segmentation loss and reconstruction loss terms. During the backward pass, the ViT encoder ($\text{ViT}_x$) stays in the CS-FE model, where it receives the input images ($x$) and extracts the corresponding ViT features ($\text{ViT}_x(x)$). The rest of the steps behave similarly, except for combining global features. The V1 version is ideal when segmentation quality is critical and some additional complexity is acceptable.

### 2.2.2 DiffusedSplitFed-V2: Latent diffusion with global feature fusion

DiffusedSplitFed-V2 applies latent diffusion with global feature fusion. During the forward pass, forward diffusion is first applied to CS-FE features ($z_{\text{FE}}$), forming noisy CS-FE features ($z_{\text{FE\_noisy}}$). $z_{\text{FE\_noisy}}$ is denoised using $\text{LDDM}_1$, generating $LF_{\text{syn\_FE}}$. The reconstruction loss at $\text{LDDM}_1$ ($l_{\text{recon1}}$) is recorded. $LF_{\text{syn\_FE}}$ are then combined with $GF_{\text{syn\_FE}}$. The combined feature set $z_{\text{OUT\_FE}}$ is sent to the SS model to carry on the forward pass. The SS output ($z_{\text{SS}}$) is then subjected to the second forward diffusion, forming noisy SS features ($z_{\text{SS\_noisy}}$). $z_{\text{SS\_noisy}}$ is denoised using $\text{LDDM}_2$, generating $LF_{\text{syn\_SS}}$. The reconstruction loss at $\text{LDDM}_2$ ($l_{\text{recon2}}$) is recorded. $LF_{\text{syn\_SS}}$ are then combined with $GF_{\text{syn\_SS}}$. The combined feature set $z_{\text{OUT\_SS}}$ is sent to the BE model to complete the forward pass. The final predictions ($\hat{y}$) are obtained, segmentation loss is recorded ($l_{seg}(\hat{y}, y)$), and optimization continues, minimizing the combined loss function of the segmentation loss and reconstruction loss terms. The same architecture in reverse is used during the backward pass, except for combining global features. The V2 version is suited for scenarios where ViT computation is costly or label/ image access is limited at split points during training and where a balance between strong privacy and moderate utility is desired.

### 2.2.3 DiffusedSplitFed-V3: lightweight latent diffusion

DiffusedSplitFed-V3 solely applies latent diffusion in the split model. During the forward pass, forward diffusion is first applied to CS-FE features ($z_{\text{FE}}$), forming $z_{\text{FE\_noisy}}$. $z_{\text{FE\_noisy}}$ is denoised using $\text{LDDM}_1$, generating $LF_{syn\_FE}$. The reconstruction loss at $\text{LDDM}_1$ ($l_{\text{recon1}}$) is recorded. $LF_{\text{syn\_FE}}$ is then sent to the SS model to carry on the forward pass. The SS output ($z_{\text{SS}}$) is then subjected to the second forward diffusion, forming $z_{\text{SS\_noisy}}$. $z_{\text{SS\_noisy}}$ is denoised using $\text{LDDM}_2$, generating $LF_{syn\_SS}$. The reconstruction loss at the $\text{LDDM}_2$ ($l_{\text{recon2}}$) is recorded. $LF_{\text{syn\_SS}}$ is then sent to the BE model to complete the forward pass. The same architecture in reverse is used during the backward pass. The V3 version is ideal for resource-constrained and high-privacy scenarios, where minimal architectural complexity is preferred.

## 2.3 Feature fusion strategies

We explored two distinct strategies during feature fusion: weighted feature fusion and attention-based feature fusion. The weighted feature fusion approach computes attention weights along the channel dimension, enabling the model to compute the contribution of each channel when combining feature maps. Attention-based feature fusion uses the compressai's Bégaint et al. (2020) attention block, which employs self-attention mechanisms to learn dynamic weights across both spatial and channel dimensions. This enables the network to focus on key regions and channels, yielding a more relevant feature representation for further processing. Details are listed in Appendix D.

## 3 Performance evaluation

### 3.1 Experimental setup

We conducted experiments on three medical imaging datasets introduced in Section 1: (1) **HAM10K** Tschandl et al. (2018); Tschandl (2018), comprising 10,015 dermatoscopic RGB images with binary ground truth masks for skin lesion and background segmentation; (2) **Blastocysts** Lockhart et al. (2019), with 781 RGB human day-5 embryo images annotated into five classes: zona pellucida (ZP), trophectoderm (TE), blastocoel (BL), inner cell mass (ICM), and background (BG); and (3) **FHPsAOPMSB** Lu et al. (2022); Kuş & Aydin (2024), containing 4,000 Intrapartum Transperineal Ultrasound (ITU) images segmented into fetal head, pubic symphysis, and background. Samples from each dataset were randomly distributed across five clients with uneven splits: HAM10K (6325, 241, 71, 2359, 9), Blastocysts (240, 120, 85, 179, 87), and FHPsAOPMSB

---

**Algorithm 1** DiffusedSplitFed-V1 Training procedure during the forward pass

---

**Input:** Global split model $w = \{w_{\text{FE}}, w_{\text{SS}}, w_{\text{BE}}\}$, LDDMs $\theta_1, \theta_2$, global features $GF_{\text{syn\_FE}}$, $GF_{\text{syn\_SS}}$, ViT label encoder $\text{ViT}_y$

**Output:** Updated $w, \theta_1, \theta_2$.

 1: **for** each communication round $t = 1$ to $T$ **do**
 2:  **for** each selected client $i$ **in parallel do**
 3:    **for** local epoch $e = 1$ to $E$ **do**
 4:      **for** each batch $(x, y) \in$ client $i$'s data **do**
 5:        $z_{\text{FE}} \leftarrow \text{FE\_model}(x; w_{\text{FE}})$
 6:        $z_{\text{vit\_noisy}} \leftarrow \text{ForwardDiffusion}(\text{ViT}_y(y))$
 7:        $z_{\text{combined\_FE}} \leftarrow \text{Combine}(z_{\text{FE}}, z_{\text{vit\_noisy}})$
 8:        $LF_{\text{syn\_FE}} \leftarrow \text{LDDM}_1(z_{\text{combined\_FE}}; \theta_1)$
 9:        $\ell_{\text{recon1}} \leftarrow \text{MSE}(z_{\text{FE}}, LF_{\text{syn\_FE}})$
10:        $z_{\text{OUT\_FE}} \leftarrow \text{Combine}(LF_{\text{syn\_FE}}, GF_{\text{syn\_FE}})$
11:        $z_{\text{SS}} \leftarrow \text{SS\_model}(z_{\text{OUT\_FE}}; w_{\text{SS}})$
12:        $z_{\text{combined\_SS}} \leftarrow \text{Combine}(z_{\text{SS}}, z_{\text{vit\_noisy}})$
13:        $LF_{\text{syn\_SS}} \leftarrow \text{LDDM}_2(z_{\text{combined\_SS}}; \theta_2)$
14:        $\ell_{\text{recon2}} \leftarrow \text{MSE}(z_{\text{server\_out}}, LF_{\text{syn\_SS}})$
15:        $z_{\text{OUT\_SS}} \leftarrow \text{Combine}(LF_{\text{syn\_SS}}, GF_{\text{syn\_SS}})$
16:        $\hat{y} \leftarrow \text{BE\_model}(z_{\text{OUT\_SS}}; w_{\text{BE}})$
17:        Compute loss: $\ell = \ell_{\text{seg}}(\hat{y}, y) + \ell_{\text{recon1}} + \ell_{\text{recon2}}$
18:        Update $w, \theta_1, \theta_2$ using gradients
19:      **end for**
20:    **end for**
21:  **end for**
22:  Server aggregates the updated local models, LDDMs, and ViT label encoders from all clients
23: **end for**
24: **Return:** Updated model $w$ and LDDMs $\theta_1, \theta_2$

---

**Algorithm 2** DiffusedSplitFed-V2 Training procedure during the forward pass

---

**Input:** Global split model $w = \{w_{\text{FE}}, w_{\text{SS}}, w_{\text{BE}}\}$, LDDMs $\theta_1, \theta_2$, global features $GF_{\text{syn\_FE}}$, $GF_{\text{syn\_SS}}$

**Output:** Updated $w, \theta_1, \theta_2$

 1: **for** each communication round $t = 1$ to $T$ **do**
 2:  **for** each selected client $i$ **in parallel do**
 3:    **for** local epoch $e = 1$ to $E$ **do**
 4:      **for** each batch $(x, y) \in$ client $i$'s data **do**
 5:        $z_{\text{FE}} \leftarrow \text{FE\_model}(x; w_{\text{FE}})$
 6:        $z_{\text{FE\_noisy}} \leftarrow \text{ForwardDiffusion}(z_{\text{FE}})$
 7:        $LF_{\text{syn\_FE}} \leftarrow \text{LDDM}_1(z_{\text{FE\_noisy}}; \theta_1)$
 8:        $\ell_{\text{recon1}} \leftarrow \text{MSE}(z_{\text{FE}}, LF_{\text{syn\_FE}})$
 9:        $z_{\text{OUT\_FE}} \leftarrow \text{Combine}(LF_{\text{syn\_FE}}, GF_{\text{syn\_FE}})$
10:        $z_{\text{SS}} \leftarrow \text{Server\_model}(z_{\text{OUT\_FE}}; w_{\text{SS}})$
11:        $z_{\text{SS\_noisy}} \leftarrow \text{ForwardDiffusion}(z_{\text{SS}})$
12:        $LF_{\text{syn\_SS}} \leftarrow \text{LDDM}_2(z_{\text{SS\_noisy}}; \theta_2)$
13:        $\ell_{\text{recon2}} \leftarrow \text{MSE}(z_{\text{SS}}, LF_{\text{syn\_SS}})$
14:        $z_{\text{OUT\_SS}} \leftarrow \text{Combine}(LF_{\text{syn\_SS}}, GF_{\text{syn\_SS}})$
15:        $\hat{y} \leftarrow \text{BE\_model}(z_{\text{OUT\_SS}}; w_{\text{BE}})$
16:        Compute loss: $\ell = \ell_{\text{seg}}(\hat{y}, y) + \ell_{\text{recon1}} + \ell_{\text{recon2}}$
17:        Update $w, \theta_1, \theta_2$ using gradients
18:      **end for**
19:    **end for**
20:  **end for**
21:  Server aggregates the updated local models and LDDMs from all clients
22: **end for**
23: **Return:** Updated model $w$ and LDDMs $\theta_1, \theta_2$

---

**Algorithm 3** DiffusedSplitFed-V3 Training procedure during the forward pass

**Input:** Global split model $w = \{w_{\text{FE}}, w_{\text{SS}}, w_{\text{BE}}\}$, LDDMs $\theta_1, \theta_2$
**Output:** Updated $w, \theta_1, \theta_2$

1: **for** each communication round $t = 1$ to $T$ **do**
2:     **for** each selected client $i$ **in parallel do**
3:         **for** local epoch $e = 1$ to $E$ **do**
4:             **for** each batch $(x, y) \in$ client $i$'s data **do**
5:                 $z_{\text{FE}} \leftarrow \text{FE\_model}(x; w_{\text{FE}})$
6:                 $z_{\text{FE\_noisy}} \leftarrow \text{ForwardDiffusion}(z_{\text{FE}})$
7:                 $LF_{\text{syn\_FE}} \leftarrow \text{LDDM}_1(z_{\text{FE\_noisy}}; \theta_1)$
8:                 $\ell_{\text{recon1}} \leftarrow \text{MSE}(z_{\text{FE}}, LF_{\text{syn\_FE}})$
9:                 $z_{\text{SS}} \leftarrow \text{Server\_model}(LF_{\text{syn\_FE}}; w_{\text{SS}})$
10:                 $z_{\text{SS\_noisy}} \leftarrow \text{ForwardDiffusion}(z_{\text{SS}})$
11:                 $LF_{\text{syn\_SS}} \leftarrow \text{LDDM}_2(z_{\text{SS\_noisy}}; \theta_2)$
12:                 $\ell_{\text{recon2}} \leftarrow \text{MSE}(z_{\text{SS}}, LF_{\text{syn\_SS}})$
13:                 $\hat{y} \leftarrow \text{BE\_model}(LF_{\text{syn\_SS}}; w_{\text{BE}})$
14:                 Compute loss: $\ell = \ell_{\text{seg}}(\hat{y}, y) + \ell_{\text{recon1}} + \ell_{\text{recon2}}$
15:                 Update $w, \theta_1, \theta_2$ using gradients
16:             **end for**
17:         **end for**
18:     **end for**
19:     Server aggregates the updated local models and LDDMs from all clients
20: **end for**
21: **Return:** Updated model $w$ and LDDMs $\theta_1, \theta_2$

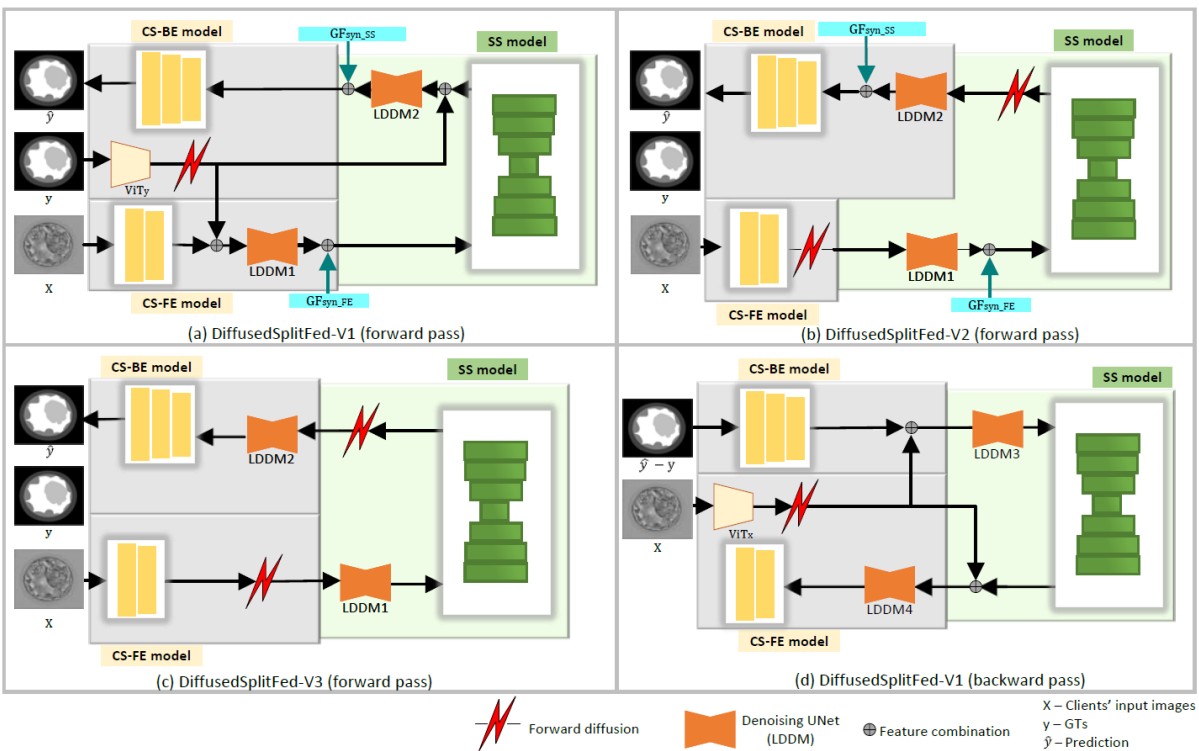

Figure 2: Proposed DiffusedSplitFed variants: (a) DiffusedSplitFed-V1 (forward pass), (b) DiffusedSplitFed-V2 (forward pass), (c) DiffusedSplitFed-V3 (forward pass), (d) DiffusedSplitFed-V1 backward pass).

(343, 914, 1143, 57, 743). Each client used 85% of its data for training and 15% for validation. Test set sizes were 70, 1010, and 800, respectively. All samples were resized to $240 \times 240$ pixels. Data augmentation included flipping, rotation, RGB shifts, normalization, and brightness or contrast adjustments. We used Soft Dice Loss Sudre et al. (2017) for training and average Intersection over Union (IoU) Cox & Cox (2008) for evaluation. For Blastocysts, IoU was averaged over TE, ZP, BL, and ICM; for FHPsAOPMSB, over fetal head and pubic symphysis.

Forward noise increased linearly from $\beta_1 = 0.0001$ to $\beta_T = 0.02$ over $T = 1000$ diffusion steps Ho et al. (2020). The denoising UNet (Fig. 5) incorporates sinusoidal time embeddings to encode timestep $t$, aiding noise-level inference and improving stability. This is especially beneficial in SplitFed scenario with asynchronous updates. We used the Adam optimizer with a learning rate of 0.0001, training for 12 local epochs over 10 communication rounds. Experiments were conducted on the x, y, and z clusters using high-performance computing resources provided by ABC[1]. We used a simple Linux utility for resource management scripts (SLURM) to request 1 GPU, 8 CPU cores, and 64 GB RAM per experiment on a single node.

## 3.2 DiffusedSplitFed performance

Table 1 presents the segmentation IoU scores for the Baseline SplitFed and the three DiffusedSplitFed variants—each evaluated under two settings: applying diffusion during the forward pass only (FP) and applying diffusion during both the forward and backward passes (FBP)—across all three datasets. Nearly all DiffusedSplitFed variants surpass the Baseline SplitFed performance, as highlighted in bold, demonstrating the effectiveness of integrating LDDMs in preserving utility while offering enhanced privacy. At the same time, FBP maintains comparable segmentation performance to FP while further strengthening privacy, showing that applying diffusion at both forward and backward introduces negligible performance degradation. The V3 variant achieves better segmentation performance without relying on ViT-based label conditioning or global feature fusion, suggesting that lightweight latent diffusion alone is a powerful privacy-preserving mechanism. However, the V2 and V3 variants serve to explore the directions for privacy-utility trade-offs. The V1 variant is suited for high-fidelity structure preservation that is facilitated by the ViT, and scenarios where label embeddings are meaningful, while the V2 variant balances strong privacy guarantees with global feature context, without label priors. The V3 variant can be optimized for lightweight, high-privacy deployments with minimal architectural complexity.

We also observed some dataset-specific trends–HAM10K has strong improvement across all variants and settings. FHPsAOPMSB shows small but consistent improvement across all variants, especially in V3. Blastocysts results are mixed-—while V1 lags behind the baseline in both FP and FBP, V3 surpasses the baseline, particularly in the FP setting. This suggests that V3 is more robust for the Blastocysts dataset. These trends might be justified along with the differences in dataset characteristics. HAM10K is a simple binary segmentation task involving natural RGB skin lesion images, with a relatively larger sample size of 10,015 images. Its simplicity and sample volume might have contributed to more stable training and generalization, enabling all DiffusedSplitFed variants to yield better performance. In contrast, FHPsAOPMSB presents a more challenging 3-class segmentation task on ITU images, which are harder to segment due to low contrast and structural ambiguity. The sample size is also comparatively small, comprising 4,000 images, which can limit the performance improvements. Segmentation of Blastocysts is even harder, involving 5-class segmentation on microscopic embryo images with only 781 samples. The high granularity of classes and limited data explain why the DiffusedSplitFed variants keep struggling to demonstrate comparatively better performance, except in V3.

## 3.3 Attacks and Defense evaluation

We evaluate the privacy-preserving capabilities of DiffusedSplitFed framework through both quantitative and qualitative analyses. We simulate adversarial reconstruction attacks, where a malicious attacker attempts to recover original images from intermediate features. We also perform a qualitative visual inspection of the feature and gradient maps transmitted across split points. These evaluations are designed to assess the extent to which sensitive semantic information is leaking through intermediate representations during split model

---

[1] omitted to preserve anonymity

training. By comparing the Baseline SplitFed with the three variants of DiffusedSplitFed, we demonstrate how our diffusion-based feature perturbation and denoising strategy mitigates reconstruction risks, preserving local data privacy without sacrificing segmentation performance.

### 3.3.1 Adversarial reconstruction for privacy evaluation

To empirically validate the privacy guarantees of DiffusedSplitFed, we simulate passive attacker models to reconstruct the original input images from the intermediate features transmitted at the CS-FE to the server split point. Since the attacker does not interfere with the segmentation task, we assess privacy leakage based on the fidelity of the reconstructed inputs–more accurate reconstructions imply greater leakage. Thus, a robust privacy-preserving framework should reduce the attacker's reconstruction quality. We simulate two attacker models depending on dataset complexity and image modality:

1. HAM10K (dermoscopic images): We employ a simple UNet-based autoencoder as the attacker. This model takes the intermediate feature map as input and is trained to reconstruct the original RGB images. The attacker is optimized using Mean Squared Error (MSE) Bickel & Doksum (2015) between the reconstructed and original images. This simple UNet architecture is adequate for HAM10K, given its binary segmentation task and the dataset's relatively uniform image structure.

2. Blastocysts and FHPsAOPMSB (microscopic and ultrasound images): Given the higher complexity of these datasets—featuring fine-grained biological details and multi-class segmentation—we employ a more expressive CNN-Transformer hybrid attacker (see fig. 6 in Appendix B). This model captures both local and global dependencies in the feature maps, improving reconstruction fidelity for complex structures.

All attacker models are trained in a fully supervised manner, with access to the original images during training. The goal is to reconstruct the original input from immediate features to simulate a reconstruction attack. Evaluation is conducted using standard reconstruction metrics: Peak Signal-to-Noise Ratio (PSNR) Wolf & Pinson (2009), Structural Similarity Index Measure (SSIM) Wang et al. (2004), Fréchet Inception Distance (FID) Heusel et al. (2017), and MSE. Higher PSNR and SSIM indicate better reconstruction quality (and thus worse privacy), whereas higher FID and MSE suggest degraded perceptual and pixel-wise fidelity (and therefore better privacy protection). Table 2 presents the quantitative results, revealing that DiffusedSplitFed variants— V1, V2 (and V3)—significantly hinder the attacker's ability to reconstruct the input, highlighting the effectiveness of feature perturbation and diffusion in preserving privacy.

Table 1: Segmentation performance (mean IoU) for Baseline SplitFed and DiffusedSplitFed variants. **BS:** Baseline SplitFed, **DS:** DiffusedSplitFed. **FP:** Forward pass, **FBP:** Forward and backward pass. Bold values indicate cases where DS variants surpass BS.

| Dataset | BS | DS-V1 | | DS-V2 | | DS-V3 | |
|---|---|---|---|---|---|---|---|
| | | FP | FBP | FP | FBP | FP | FBP |
| HAM10K | 0.892 | **0.897** | **0.912** | **0.894** | **0.892** | **0.911** | **0.908** |
| Blastocyst | 0.892 | 0.851 | 0.845 | 0.850 | 0.870 | **0.902** | **0.892** |
| FHPsAOPMSB | 0.886 | **0.887** | 0.860 | 0.863 | 0.881 | **0.888** | **0.888** |

Table 2: Reconstruction performance (lower PSNR, SSIM; higher FID, MSE = better privacy). **BS:** Baseline SplitFed, **DS:** DiffusedSplitFed.

| Dataset | Metric | BS | DS-V1 | DS-V2 (&V3) |
|---|---|---|---|---|
| HAM10K | PSNR | 20.90 | 20.54 | 16.17 |
| | SSIM | 0.846 | 0.816 | 0.696 |
| | FID | 8517.8 | 10530.4 | 22039.8 |
| | MSE | 0.0105 | 0.0130 | 0.0272 |
| Blastocyst | PSNR | 20.15 | 19.30 | 17.33 |
| | SSIM | 0.766 | 0.600 | 0.742 |
| | FID | 6873.4 | 8809.8 | 14075.8 |
| | MSE | 0.0114 | 0.0145 | 0.0229 |
| FHPsAOPMSB | PSNR | 17.15 | 16.98 | 15.94 |
| | SSIM | 0.725 | 0.696 | 0.692 |
| | FID | 17770.2 | 17920.6 | 18433.6 |
| | MSE | 0.0226 | 0.0228 | 0.0238 |

### 3.3.2 Visual and quantitative assessment of privacy leakage

Table 8 presents a qualitative comparison of privacy leakage across Baseline SplitFed and the DiffusedSplitFed variants. We show the visual feature maps and quantitative reconstruction metrics calculated at the CS-FE

to server split point for each dataset during the first epoch of clients' local training. While Baseline SplitFed transmits raw intermediate features, both DiffusedSplitFed-V1 and V2 (similarly V3) transmit obfuscated and noisy representations, reducing their vulnerability to reconstruction or inversion attacks. In Section 3.3.1, it is noticeable that the reconstructed outputs for DiffusedSplitFed variants show more distortion and less semantic fidelity than those for Baseline SplitFed, visually suggesting reduced information leakage. Table 8 (Appendix B), provides a qualitative assessment of local privacy at the CS-FE to server split point by presenting intermediate feature maps, along with reconstruction metrics for DiffusedSplitFed-V1 (Column 2) and DiffusedSplitFed-V2 & V3 (Column 3), each computed with respect to Baseline SplitFed (Column 1). Across the three datasets, DiffusedSplitFed variants consistently yield higher MSE, SD, and FID scores and lower PSNR and SSIM values, further indicating that the reconstructions are less similar to the original inputs. This supports that DiffusedSplitFed enhances local privacy at the client-server communication through our three architectural designs.

### 3.4 Privacy–utility trade-off analysis

Our privacy-utility trade-off analysis, summarized in Table 3, illustrates how each variant of DiffusedSplitFed balances segmentation performance (utility) with resistance to reconstruction attacks (privacy). This evaluation identifies more effective variants in achieving high utility while minimizing privacy leakage. As seen in Table 2 Baseline SplitFed achieves the best reconstruction performance with attacker models (the highest PSNR, SSIM, and lower FID and MSE scores). However, this comes at the cost of privacy, making it easier for an attacker to reconstruct the original images. In contrast, DiffusedSplitFed-V2 and V3 demonstrate the strongest privacy preservation (lower PSNR, SSIM, and higher FID and MSE scores), indicating effective obfuscation of sensitive features. DiffusedSplitFed-V1 offers a balanced trade-off-while its PSNR and FID are slightly lower than the baseline, its SSIM remains comparable, suggesting some structural fidelity is preserved. This balance shows that it effectively preserves privacy while enhancing segmentation performance. DiffusedSplitFed-V1 exhibits higher complexity than Baseline SplitFed, while V2 and V3 show only a slight increase in trainable parameters and floating-point operations (FLOPS).

Table 3: Qualitative trade-offs in terms of privacy level, reconstruction quality (Recon.quality), segmentation performance, model complexity, and potential use cases. **BS:** Baseline SplitFed, **DS:** DiffusedSplitFed, **TP:** Trainable parameters, **FLOPS:** Floating-point operations in the FP (forward pass) setting.

| Scenario | Privacy | Recon. quality | Segmentation performance | Model complexity (TP & FLOPS) | Use case |
|---|---|---|---|---|---|
| BS | Worst | Best | Moderate | 7.76M & 13.77GMAC | When privacy not a priority |
| DS-V1 | Moderate | Moderate | Better w/ large datasets | 100.19M & 32.12GMAC | Balanced for real-world use |
| DS-V2 | Strongest | Weakest | Better w/ large datasets | 13.62M & 15.25GMAC | High-privacy use |
| DS-V3 | Strongest | Weakest | Better w/ all datasets | 13.62M & 15.25GMAC | High-privacy use |

## 4 Comparison with the SoTA

### 4.1 Comparison with SoTA latent diffusion approaches

For comparison, we reimplemented two SoTA baseline methods: Generative Medical Segmentation (GMS) Huo et al. (2024) and LDSeg Zaman et al. (2024). GMS is a generative framework for medical image segmentation that models the relationship between image and mask representation in the latent space. It utilizes a pretrained ViT-based image encoder and decoder, along with a latent mapping network that transforms the latent features into corresponding segmentation masks. Following GMS Huo et al. (2024), we used a learning rate of 0.002 and trained centrally for 150 epochs. In the SplitFed scenario, each client trained locally for 15 epochs per round over 10 communication rounds with uniformly distributed data. LDSeg is a latent diffusion-based segmentation framework that maps input images and masks into a shared latent space using dual ResUnet encoders (without skip connections). Forward diffusion is applied to the mask embedding, and a conditional denoising network reconstructs it based on the image embedding, followed by a label decoder.

As our datasets were not used in the original LDSeg implementation, we adopted the training parameters in the GMS paper for a fair comparison. Our proposed DiffusedSplitFed variants achieve higher IoUs than the Baseline SplitFed versions of both GMS and LDSeg, as shown in Table 4. Additionally, both our centralized and Baseline SplitFed models outperform GMS and LDSeg in terms of IoU. The corresponding cells are bolded.

Table 4: Comparison of segmentation performance (mean IoU) for centralized and SplitFed training using SoTA methods with DiffusedSplitFed variants. **C**: Centralized scenario, **BS**: Baseline SplitFed. **DS**: DiffusedSplitFed, **FP**: Latent diffusion in the forward pass, **FBP**: Latent diffusion in both the forward & backward passes. Dashes (−): configurations that are not applicable. Bold values indicate the cases where centralized or BS outperforms the considered SoTA methods.

| Dataset | Model | C | BS | DS-V1 | | DS-V2 | | DS-V3 | |
|---|---|---|---|---|---|---|---|---|---|
| | | | | FP | FBP | FP | FBP | FP | FBP |
| HAM10K | GMS | 0.896 | 0.852 | – | – | – | – | – | – |
| | LDSeg | 0.819 | 0.804 | – | – | – | – | – | – |
| | Our UNet | **0.898** | **0.897** | **0.888** | **0.882** | **0.899** | **0.889** | **0.898** | **0.896** |
| Blastocyst | GMS | 0.376 | 0.227 | – | – | – | – | – | – |
| | LDSeg | 0.685 | 0.674 | – | – | – | – | – | – |
| | Our UNet | **0.890** | **0.892** | **0.854** | **0.857** | **0.861** | **0.864** | **0.901** | **0.894** |
| FHPsAOPMSB | GMS | 0.864 | 0.844 | – | – | – | – | – | – |
| | LDSeg | 0.791 | 0.747 | – | – | – | – | – | – |
| | Our UNet | **0.888** | **0.886** | **0.891** | **0.895** | **0.800** | **0.871** | **0.888** | **0.887** |

## 4.2 Comparison with SoTA privacy-preserving approaches

We consider several widely studied privacy-preserving techniques in federated and distributed learning: (1) Differential privacy, which adds calibrated noise to features or gradients Abadi et al. (2016); Wei et al. (2020); (2) Homomorphic encryption that enables secure computation without noise but with overhead Phong et al. (2018); Xie et al. (2024); Yan et al. (2024); (3) Secure multiparty computation that performs joint computation while preserving input privacy Liu et al. (2024); Lindell (2020); and (4) Adversarial feature transformation, which obfuscates representations to hide private information Li et al. (2021); Yue et al. (2023). Among these, we implemented differential privacy at the split points in the Baseline SplitFed setting for $\sigma \in \{0.005, 0.02, 0.05, 0.2, 0.3\}$, and $Clip \in \{1, 2\}$ to fairly evaluate the performance against our proposed DiffusedSplitFed method. Table 5 shows the results, each showing a lesser mean IoU compared to the DiffusedSplitFed metrics shown in Table 1. Table 6 compares these methods in terms of privacy, cost, and complexity. Our DiffusedSplitFed variants (V1-V3) show favorable trade-offs with high privacy, low-to-medium computational cost, and low model complexity.

Table 5: Segmentation performance (Mean IoU) for Baseline SplitFed with differential privacy (DP) noise. **FP$_{DP}$**: DP noise applied in the forward pass, **FBP$_{DP}$**: DP noise applied in both forward and backward passes, $\sigma$: standard deviation of the Gaussian noise added for DP, $Clip$: gradient clipping threshold used before noise injection.

| Dataset | Clip | FP$_{DP}$ | | | | | FBP$_{DP}$ | | | | |
|---|---|---|---|---|---|---|---|---|---|---|---|
| | | $\sigma$=0.005 | $\sigma$=0.02 | $\sigma$=0.05 | $\sigma$=0.2 | $\sigma$=0.3 | $\sigma$=0.005 | $\sigma$=0.02 | $\sigma$=0.05 | $\sigma$=0.2 | $\sigma$=0.3 |
| HAM10K | 1 | 0.796 | 0.780 | 0.696 | 0.669 | 0.350 | 0.801 | 0.780 | 0.764 | 0.347 | 0.158 |
| | 2 | 0.801 | 0.792 | 0.781 | 0.701 | 0.663 | 0.791 | 0.667 | 0.511 | 0.113 | 0.035 |
| Blastocyst | 1 | 0.855 | 0.752 | 0.607 | 0.285 | 0.261 | 0.855 | 0.384 | 0.113 | 0.070 | 0.063 |
| | 2 | 0.861 | 0.508 | 0.498 | 0.289 | 0.279 | 0.841 | 0.252 | 0.089 | 0.053 | 0.066 |
| FHPsAOPMSB | 1 | 0.878 | 0.844 | 0.753 | 0.669 | 0.628 | 0.881 | 0.846 | 0.752 | 0.664 | 0.635 |
| | 2 | 0.852 | 0.822 | 0.724 | 0.628 | 0.557 | 0.878 | 0.876 | 0.137 | 0.040 | 0.036 |

Table 6: Comparison of privacy-preserving methods in terms of privacy, computational cost, and model complexity.

| Method | Privacy | Computational cost | Model complexity |
|---|---|---|---|
| Differential privacy | Medium | Low | Low |
| Homomorphic encryption | Very high | High | High |
| Secure aggregation | High | Medium | High |
| Feature shuffling and permutation | Medium | Low | Low |
| Adversarial feature transformation/ obfuscation | High | Medium | High |
| Low-rank adaptation against membership inference attacks | Medium | Low | Low |
| DiffusedSplitFed-V1 | Medium | Low/ Medium | Low |
| DiffusedSplitFed-V2 (& V3) | High | Low/ Medium | Low |

## 5 Convergence analysis overview

Our theoretical convergence analysis shows that DiffusedSplitFed variants converge to a bounded solution under standard assumptions. Full details are provided in Appendix C.

Further, we analyze the difference in the convergence bounds of the proposed DiffusedSplitFed with SoTA privacy-preserving methods, particularly in the distributed training setting, as shown in Table 7. The derivations related to the DiffusedSplitFed convergence bounds are presented in Appendix C. As visible in the bounds, differential privacy offers provable privacy. However, added noise scales inversely with the privacy budget $\varepsilon$, meaning stricter privacy (smaller $\varepsilon$) slows convergence. In addition, the quadratic dependence on $T$ in the $\mathcal{O}(\frac{T^2}{\varepsilon^2})$ term can dominate in long training regimes. Homomorphic encryption does not modify the data or gradients, preserving privacy and convergence dynamics. However, the quantization and gradient clipping terms have introduced small errors, which are captured in the second term of the bound. Also, homomorphic encryption incurs computational and communication overhead, although it is not reflected in the bound. Compared to others, the DiffusedSplitFed bound allows for tight control over bias through the quality of denoising $\epsilon_d^{(j)}$. In addition, the use of LDDMs enables feature-level perturbation and reconstruction, preserving privacy without explicit encryption or additive noise.

## 6 Conclusion

In this paper, we introduced DiffusedSplitFed, the first SplitFed framework that integrates LDDMs with dual conditioning and global feature fusion, applied to both forward and backward passes to obfuscate latent features for privacy preservation. Our architecture enhances the traditional SplitFed by introducing diffusion, denoising, and feature perturbation, mitigating reconstruction attacks and information leakage while preserving privacy. We designed and evaluated three variants of DiffusedSplitFed to explore the privacy-utility-complexity trade-offs in SplitFed segmentation. Extensive experiments on three diverse medical imaging datasets—HAM10K, Blastocysts, and FHPsAOPMSB demonstrated that DiffusedSplitFed outperforms Baseline SplitFed and SoTA generative baselines in terms of segmentation performance while preserving privacy. To the best of our knowledge, this is the first work to incorporate latent diffusion into SplitFed learning for aiming for privacy enhancement, offering a generative paradigm for collaborative medical AI.

**Author Contributions**

Omitted to preserve anonymity.

**Acknowledgments**

The authors thank xxx for their funding support (omitted to preserve anonymity).

Table 7: Convergence bound variation of privacy-preserving methods in distributed training.

| Method | Convergence bound effect |
|---|---|
| Differential privacy Wei et al. (2020) | The expected convergence upper bound, for a fixed privacy level $\varepsilon$ and after $T$ global aggregation rounds:

$$\mathbb{E}\left[F\left(\tilde{w}^{(T)}\right) - F(w^*)\right] \leq P^T \Delta + \left(\frac{\kappa_1 T}{\varepsilon} + \frac{\kappa_0 T^2}{\varepsilon^2}\right)\left(1 - P^T\right)$$

where:
• $\Delta = F(w^{(0)}) - F(w^*)$ is the initial optimality gap,
• $P = 1 + 2l\lambda_2$ (with $\lambda_2 < 0$) determines the exponential decay rate,
• $\kappa_1$ and $\kappa_0$ are constants that depend on gradient norms, Lipschitz smoothness constants, and the noise scale.
• $\varepsilon$ is the privacy budget; smaller $\varepsilon$ implies stronger privacy, but slower convergence.

The precise convergence bound can be stated as:

$$\frac{1}{T}\sum_{t=0}^{T-1}\mathbb{E}[\|\nabla\mathcal{L}(w_t)\|^2] \leq \mathcal{O}\left(\frac{1}{T}\right) + \mathcal{O}\left(\frac{T}{\varepsilon} + \frac{T^2}{\varepsilon^2}\right)$$ |
| Homomorphic encryption Yan et al. (2024) | The convergence upper bound is given by:

$$\frac{1}{T}\sum_{t=0}^{T-1}\mathbb{E}\left[\|\nabla F(\theta_t)\|^2\right] \leq \underbrace{\frac{2\left[F(\theta_0) - F(\theta^*)\right]}{T\eta} + \frac{\sigma^2}{NB}}_{\text{Vanilla FL Error (E}_{\text{Vanilla FL}})} + \underbrace{\frac{dC^2}{N2^{2b}}}_{\text{Quantization Error (E}_{\text{Q}})}$$

where:
• $T$: total number of communication rounds,
• $\eta$: learning rate,
• $\sigma^2$: stochastic gradient variance,
• $B$: local batch size,
• $d$: gradient dimension,
• $C$: clipping threshold for gradients,
• $b$: number of quantization bits.

The precise convergence bound can be stated as:

$$\frac{1}{T}\sum_{t=0}^{T-1}\mathbb{E}\left[\|\nabla\mathcal{L}(w_t)\|^2\right] \leq \mathcal{O}\left(\frac{1}{T}\right) + \mathcal{O}\left(\frac{\sigma^2}{NB} + \frac{dC^2}{N2^{2b}}\right)$$ |
| DiffusedSplitFed-V1,V2 and V3 (derived in Appendix C) | The precise convergence bound can be stated as:

$$\frac{1}{T}\sum_{t=0}^{T-1}\mathbb{E}\left[\|\nabla\mathcal{L}(w_t)\|^2\right] \leq \mathcal{O}(1/T) + \mathcal{O}(\eta(\sigma^2 + \sum \epsilon_d^{(j)}))$$ |

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

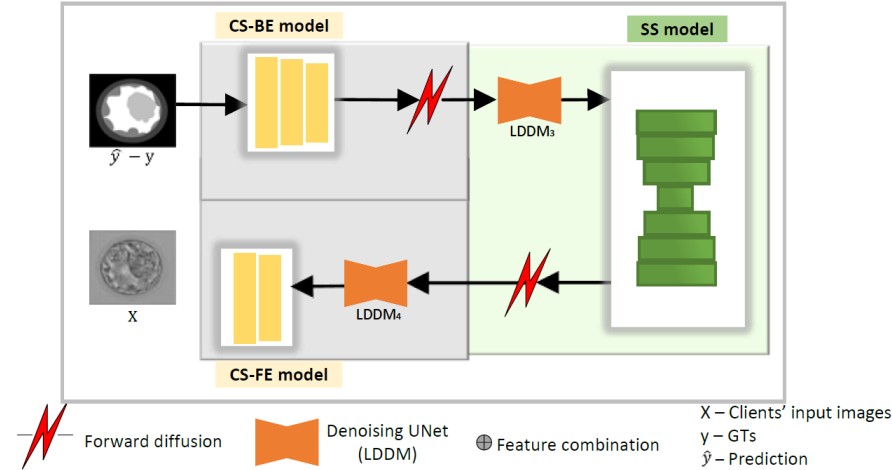

Figure 3: DiffusedSplitFed-V2 (& V3) architecture-Backward pass.

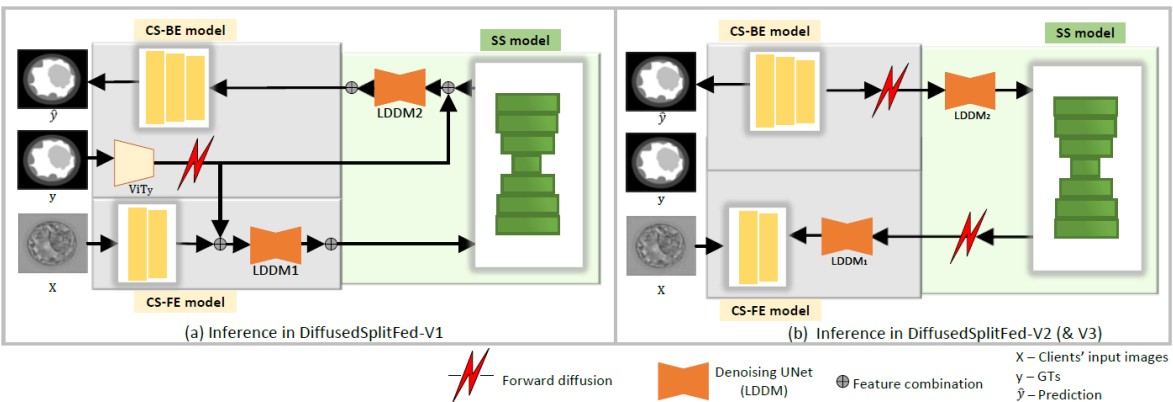

Figure 4: Inference in DiffusedSplitFed-V1 and V2 (& V3).

# A    Architectural designs-supplementary details

Fig. 3 illustrates the backward pass architectures of DiffusedSplitFed-V2 and V3. Fig. 4(a) and (b) show the architectures during the inference phase for DiffusedSplitFed-V1, and DiffusedSplitFed-V2(& V3). Fig. 5 is the denoising UNet we used. Fig. 6 is the transformer CNN hybrid architecture used in attacker model training for Blastocysts and FHPsAOPMSB data.

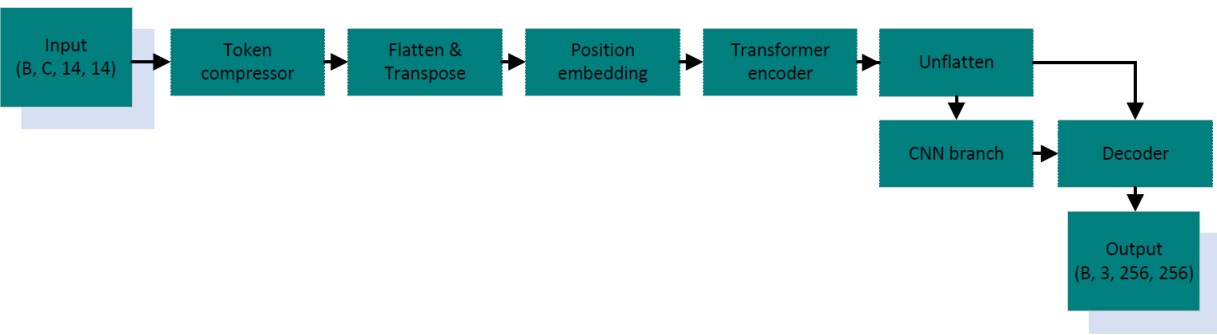

Figure 6: Transformer CNN hybrid architecture used in attacker model training for Blastocysts and FHP-sAOPMSB data.

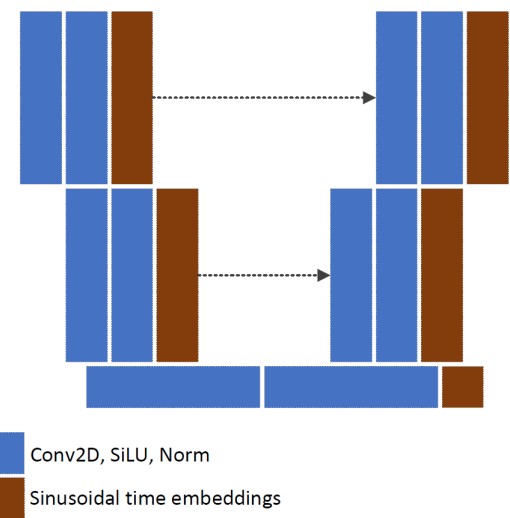

Figure 5: Denoising UNet used at the two split points during client's local training.

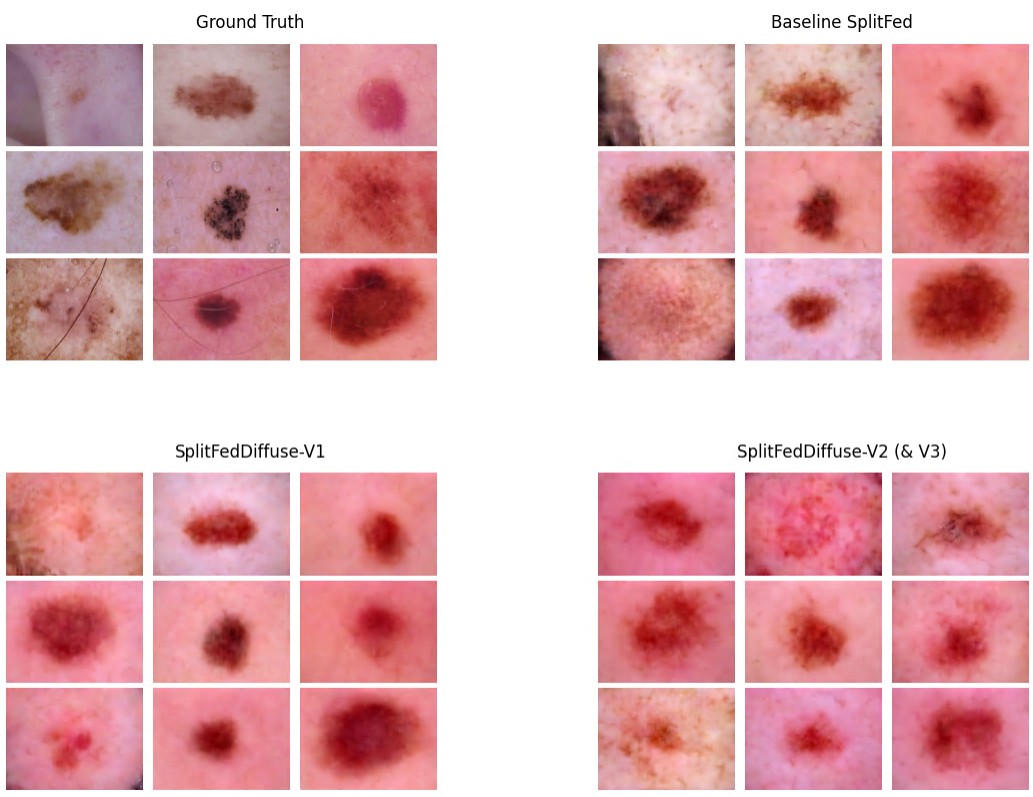

Figure 7: Predictions of the attacker model with the HAM10K.

## B   Adversarial reconstruction-supplementary details

Fig. 7, 8 and 9 illustrate the predictions of the attacker model with the HAM10K, Blastocysts, and FHPsAOPMSB datasets.

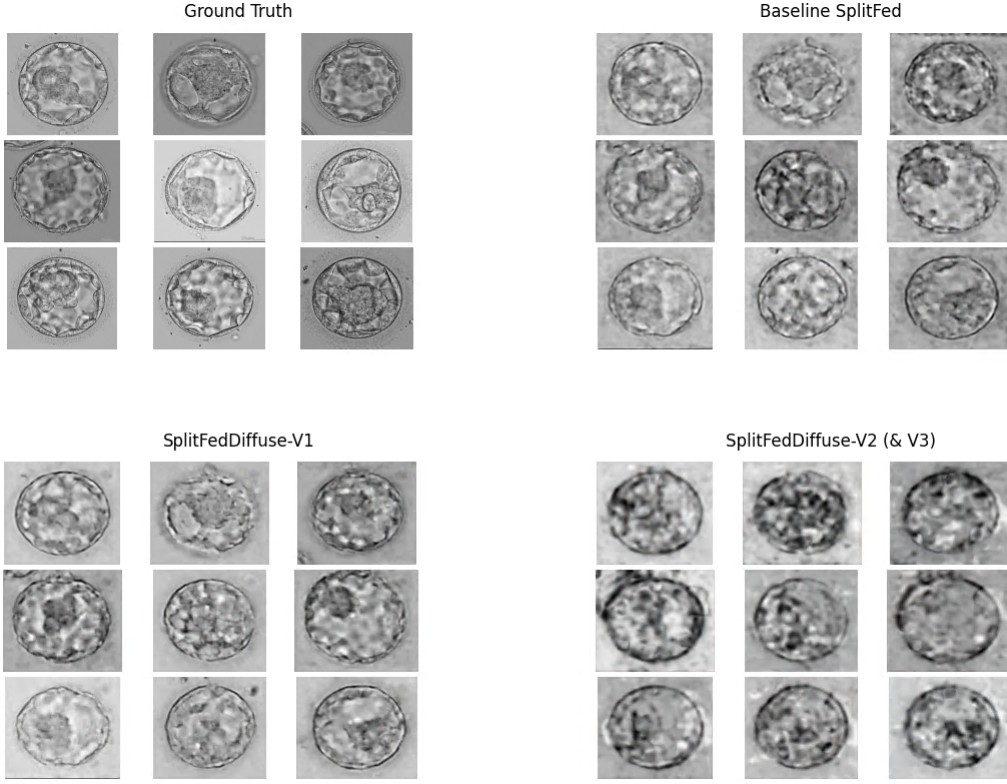

Figure 8: Predictions of the attacker model with the Blastocysts.

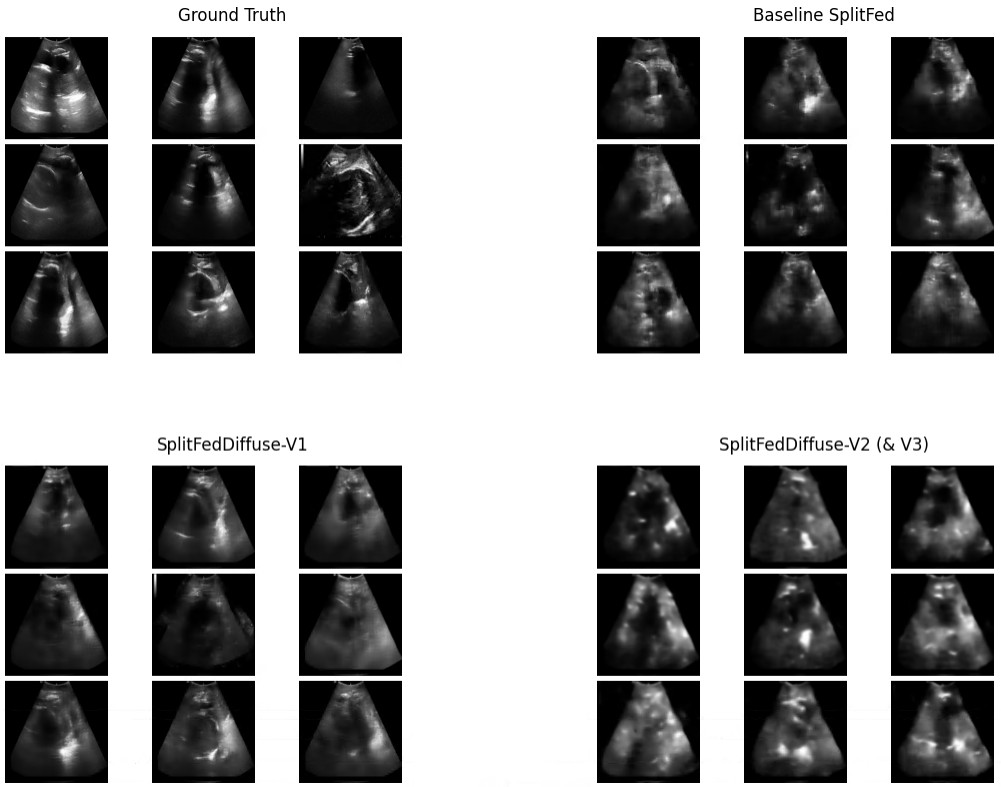

Figure 9: Predictions of the attacker model with the FHPsAOPMSB data.

Table 8: Qualitative assessment of local privacy at the CS-FE to server split point (split point 1). **BS**: Baseline SplitFed. **DS**: DiffusedSplitFed.

| Dataset | BS | DS-V1 | DS-V2 (& V3) |
|---|---|---|---|
| HAM10K | *Metric computation reference* | MSE: 1.6562
SD: 1.2429
FID: 650536.83
PSNR: 13.1550
SSIM: 0.1741 | MSE: 1.5461
SD: 1.1986
FID: 1596400.05
PSNR: 7.1069
SSIM: 0.02146 |
| Blastocysts | *Metric computation reference* | MSE: 0.8968
SD: 0.9469
FID: 697347.68
PSNR: 12.9467
SSIM: 0.2222 | MSE: 1.9997
SD: 1.4141
FID: 1600513.50
PSNR: 7.2185
SSIM: 0.0239 |
| FHPsAOPMSB | *Metric computation reference* | MSE: 1.7423
SD: 1.2799
FID: 652683.78
PSNR: 19.4899
SSIM: 0.3981 | MSE: 1.4336
SD: 1.1600
FID: 1241466.5
PSNR: 8.7040
SSIM: 0.0503 |

# C   Convergence analysis

Following convergence analysis is developed with the steps followed in Shiranthika et al. (2024). Convergence proofs related to FL Peng et al. (2025); Vora et al. (2024) and SL Wu et al. (2023) were referred to during the formulation. Prior research offers formal proofs and analytical insights into the convergence behaviour in distributed training in various settings. It came to our observation that this is the first convergence analysis related to SplitFed associated with using latent diffusion for privacy preservation and data exchange in split points.

## C.1   Problem

Let $L(w) = \frac{1}{N} \sum_{i=1}^{N} L_i(w)$ be the global objective function for the $N$ clients, where $w \in R^d$ are the model parameters. Each client performs local training using its local data, with LDDMs used at the intermediate stages in different configurations for DiffusedSplitFed-V1, DiffusedSplitFed-V2 and DiffusedSplitFed-V3.

## C.2   Assumptions

1. *L*-smoothness
   For each client $n$, $f_n(y) \le f_n(x) + \langle \nabla f(x), y - x \rangle + \frac{L}{2} \|y - x\|^2, \forall x, y$.

2. Bounded stochastic gradient square norm
   For each client $n$, $\mathbb{E}\|\nabla f_n(w)\|^2 \le \sigma^2, \forall w$.

3. Bounded reconstruction error at the split points
   Let $x_t$ be the clean/true feature at time step $t$, and let $\hat{x}_t^{(j)}$ denote the output of the $j^{th}$ LDDM, where $j \in 1, 2$ corresponds to two forward LDDMs during FP, and $j \in 1, 2, 3, 4$ corresponds to two forward LDDMs and two backward LDDMs during FBP. We assume that the feature level reconstruction error is bounded: $\mathbb{E}[\|\hat{x}_t^{(j)} - x_t\|^2] \le \epsilon_d^{(j)}$,
   and the gradient level bias is bounded: $\delta_t^{(j)} := \nabla L(w_t; \hat{x}_t^{(j)}) - \nabla L(w_t; x_t), \|\delta_t^{(j)}\| \le \beta \mathbb{E}[\|\delta_t^{(j)}\|]^2 \le \epsilon_g^{(j)} := C_j^2 . \epsilon_d^{(j)}$

4. Existence of a global optimum
   There exists at least one solution, denoted as $w^*$ to achieve the global minimum of $F(w)$.

## C.3   Lemma

Gradient bias from LDDM. Let $\tilde{g}_i(w) = \nabla L_i(w; \hat{x})$ be the gradient computed from LDDM-denoised features. Then: $\mathbb{E}\|\tilde{g}_i(w) - \nabla L_i(w)\|^2 \le \epsilon_g$. This applies to each LDDM used in each DiffusedSplitFed version's pipelines.

## C.4   Theorems

### C.4.1   Theorem 1

The convergence bound for DiffusedSplitFed-V1 is given by:
$\frac{1}{T} \sum_{t=0}^{T-1} \mathbb{E}\|\nabla L(w_t)\|^2 \le O(\frac{1}{T}) + O(\eta(\sigma^2 + \sum_{j=1}^{2} \delta_d^{(j)}))$, if 2 LDDMs are used at the split points during FP,
and
$\frac{1}{T} \sum_{t=0}^{T-1} \mathbb{E}\|\nabla L(w_t)\|^2 \le O(\frac{1}{T}) + O(\eta(\sigma^2 + \sum_{j=1}^{4} \delta_d^{(j)}))$, if 4 LDDMs are used at the split points during FBP, where $\delta_d^1$, $\delta_d^2$, $\delta_d^3$ and $\delta_d^4$ are reconstruction errors at LDDMs.

### C.4.2   Theorem 2

The convergence bound for DiffusedSplitFed-V2 is given by:
$\frac{1}{T} \sum_{t=0}^{T-1} \mathbb{E}\|\nabla L(w_t)\|^2 \le O(\frac{1}{T}) + O(\eta(\sigma^2 + \sum_{j=1}^{2} \epsilon_d^{(j)}))$, if 2 LDDMs are used at the split points during FP,
and

$\frac{1}{T}\sum_{t=0}^{T-1}\mathbb{E}\|\nabla L(w_t)\|^2 \leq O(\frac{1}{T}) + O(\eta(\sigma^2 + \sum_{j=1}^{4}\epsilon_d^{(j)}))$, if 4 LDDMs are used at the split points during FBP, where $\epsilon_d^1$, $\epsilon_d^2$,$\epsilon_d^3$ and $\epsilon_d^4$ are reconstruction errors at LDDMs.

### C.4.3  Theorem 3

The convergence bound for DiffusedSplitFed-V3 is given by:
$\frac{1}{T}\sum_{t=0}^{T-1}\mathbb{E}\|\nabla L(w_t)\|^2 \leq O(\frac{1}{T}) + O(\eta(\sigma^2 + \sum_{j=1}^{2}\zeta_d^{(j)}))$, if 2 LDDMs are used at the split points during FP, and
$\frac{1}{T}\sum_{t=0}^{T-1}\mathbb{E}\|\nabla L(w_t)\|^2 \leq O(\frac{1}{T}) + O(\eta(\sigma^2 + \sum_{j=1}^{4}\zeta_d^{(j)}))$, if 4 LDDMs are used at the split points during FBP, where $\zeta_d^1$, $\zeta_d^2$,$\zeta_d^3$ and $\zeta_d^4$ are reconstruction errors in LDDMs.

### C.4.4  Theorem 4

Perfect denoising/Baseline SplitFed All methods will reduce to the standard SGD.
The convergence bound for the Baseline SplitFed is given by:
$\frac{1}{T}\sum_{t=0}^{T-1}\mathbb{E}\|\nabla L(w_t)\|^2 \leq O(\frac{1}{T} + \eta\sigma^2)$

### C.5  Proofs of the theorems

We use assumptions 1-4.
$F(x)$ is the global loss function aggregated over clients. $F(x_t)$ is the gradient at iteration $t$. $\eta$ is the learning rate. The lemma is the key to analyzing the expected decrease in loss per communication round.

### C.5.1  Convergence bound for DiffusedSplitFed-V1

In DiffusedSplitFed-V1, we have two denoising steps (in FP), with LDDMs during the split learning process. The first LDDM is applied after combining the client's FE model output with noisy ViT features, and the second LDDM is applied after combining the server model's intermediate outputs with noisy ViT features. Both denoised outputs are then combined with global features before being passed to the next model. These two denoising operations introduce two reconstruction error terms in gradient computation.
$\delta_t^{(1)}$ is the reconstruction error of the first LDDM.
$\delta_t^{(2)}$ is the reconstruction error of the second LDDM.
Following assumption 3, we assume that each error is bounded in norm by $\beta$.
Therefore: $|\delta_t^{(1)}| \leq \beta$ and $|\delta_t^{(2)}| \leq \beta$.
The presence of global features in the feature combination will mitigate these errors, but in the analysis, we take their full effect.

The effective gradient used for the model update in the communication round $t$ is then the true gradient + LDDM-induced bias terms + stochastic noise.

$$\tilde{g}_t = \nabla F(x_t) + \delta_t^{(1)} + \delta_t^{(2)} + \epsilon_t \tag{1}$$

The global model parameters are updated by stochastic gradient descent:

$$x_{t+1} = x_t - \eta\tilde{g}_t \tag{2}$$

substituting equation (1) in equation (2):

$$x_{t+1} = x_t - \eta(\nabla F(x_t) + \delta_t^{(1)} + \delta_t^{(2)} + \epsilon_t) \tag{3}$$

Using $L-$ smoothness (Assumption 1), we bound the change in loss $F(x_{t+1}) - F(x_t)$. By the smoothness inequality applied at $x_t$ with step $-\eta\tilde{g}_t$, we have:

$$F(x_{t+1}) \leq F(x_t) + \nabla F(x_t)^T(x_{t+1} - x_t) + \frac{L}{2}\|x_{t+1} - x_t\|^2 \tag{4}$$

Substituting equation (3) into equation (4):

$$F(x_{t+1}) = F(x_t) - \eta \nabla F(x_t)^T (\nabla F(x_t) + \delta_t^{(1)} + \delta_t^{(2)} + \epsilon_t) + \frac{L}{2} \eta^2 \|\nabla F(x_t) + \delta_t^{(1)} + \delta_t^{(2)} + \epsilon_t\|^2 \quad (5)$$

In equation 5, the part $\eta \nabla F(x_t)^T (\nabla F(x_t) + \delta_t^{(1)} + \delta_t^{(2)} + \epsilon_t)$ is the inner product term, and the part $\frac{L}{2} \eta^2 \|\nabla F(x_t) + \delta_t^{(1)} + \delta_t^{(2)} + \epsilon_t\|^2$ is the squared-norm.
Taking the inner product term:

$$\nabla F(x_t)^T \tilde{g}_t = \nabla F(x_t)^T (\nabla F(x_t) + \delta_t^{(1)} + \delta_t^{(2)} + \epsilon_t) \quad (6)$$

$$\nabla F(x_t)^T \tilde{g}_t = \|\nabla F(x_t)\|^2 + \nabla F(x_t)^T (\delta_t^{(1)} + \delta_t^{(2)}) + \nabla F(x_t)^T \epsilon_t \quad (7)$$

Taking expectations conditioned on $x_t$:
Since $\epsilon_t$ is the stochastic noise, and we assume that it is unbiased:
$\mathbb{E}[\epsilon_t|x_t] = 0$.
Then the expectation of the dot product with the gradient also vanishes.
$\mathbb{E}[\nabla F(x_t)^T \epsilon_t|x_t] = 0$. Therefore,

$$\mathbb{E}[\nabla F(x_t)^T \tilde{g}_t|x_t] = \|\nabla F(x_t)\|^2 + \nabla F(x_t)^T \mathbb{E}[\delta_t^{(1)} + \delta_t^{(2)}|x_t] \quad (8)$$

$\delta_t^{(1)} + \delta_t^{(2)}$ are bias terms caused by LDDM approximations and are not zero. Therefore, we keep the inner product term $\nabla F(x_t)^T (\delta_t^{(1)} + \delta_t^{(2)})$ in the expectation.

Taking the squared-norm term:

$$\|\nabla F(x_t) + \delta_t^{(1)} + \delta_t^{(2)} + \epsilon_t\|^2 = \|\nabla F(x_t) + \delta_t^{(1)} + \delta_t^{(2)}\|^2 + 2(\nabla F(x_t) + \delta_t^{(1)} + \delta_t^{(2)})^T \epsilon_t + \|\epsilon_t\|^2 \quad (9)$$

Taking the expectation of 9 conditioning on $x_t$:

Following Assumption 2 (bounded variance), the last term becomes:

$$\mathbb{E}[\|\epsilon_t\|^2|x_t] \leq \sigma^2 \quad (10)$$

The second term becomes:

$$\mathbb{E}[(\nabla F(x_t) + \delta_t^{(1)} + \delta_t^{(2)})^T \epsilon_t|x_t] = 0 \quad (11)$$

Substituting, equation (10) and equation (11) into equation (9)'s expectation:

$$\mathbb{E}[\|\tilde{g}_t\|^2|x_t] = \|\nabla F(x_t) + \delta_t^{(1)} + \delta_t^{(2)}\|^2 + \sigma^2 \quad (12)$$

$$\mathbb{E}[\|\tilde{g}_t\|^2|x_t] = \|\nabla F(x_t)\|^2 + 2\nabla F(x_t)^T (\delta_t^{(1)} + \delta_t^{(2)}) + \|\delta_t^{(1)} + \delta_t^{(2)}\|^2 + \sigma^2 \quad (13)$$

Adding equation (8) and equation (13):

$$\begin{aligned}
\mathbb{E}[F(x_{t+1})] \leq \mathbb{E}[F(x_t)] - \eta[\|\nabla F(x_t)\|^2] + \nabla F(x_t)^T \mathbb{E}[\delta_t^{(1)} + \delta_t^{(2)}|x_t] + \\
\frac{L}{2} \eta^2 \mathbb{E}[\|\nabla F(x_t)\|^2 + 2\nabla F(x_t)^T (\delta_t^{(1)} + \delta_t^{(2)}) + \|\delta_t^{(1)} + \delta_t^{(2)}\|^2] + \frac{L}{2} \eta^2 \sigma^2
\end{aligned} \quad (14)$$

Now we bound the terms with the error terms.
By triangle inequality, $\|\delta_t^{(1)} + \delta_t^{(2)}\|^2 \leq (\|\delta_t^{(1)}\| + \|\delta_t^{(2)}\|)^2 \leq (\beta + \beta)^2 = 4\beta^2$

$$\mathbb{E}[F(x_{t+1})] \leq \mathbb{E}[F(x_t)] - \eta(1 - \frac{L\eta}{2}) \mathbb{E}[\|\nabla F(x_t)\|^2] - \eta(1 - L\eta) \mathbb{E}[\nabla F(x_t)^T (\delta_t^{(1)} + \delta_t^{(2)})] + 2L\eta^2 \beta^2 + L\frac{\eta^2}{2} \sigma^2 \quad (15)$$

Telescoping sum over $T$ communication rounds:

$$\mathbb{E}[F(x_T)] - \mathbb{E}[F(x_0)] \leq -\eta(1 - \frac{L\eta}{2}) \sum_{t=0}^{T-1} \mathbb{E}[\|\nabla F(x_t)\|^2] - \eta(1 - L\eta) \sum_{t=0}^{T-1} \mathbb{E}[\Delta_t] + 2L\eta^2 \beta^2 T + L\frac{\eta^2}{2} \sigma^2 T \quad (16)$$

where $\Delta_t := F(x_t)^T[\delta_t^{(1)} + \delta_t^{(2)}]$

According to Assumption 4, there exists an optimal solution. At convergence, $F(x_T) = F(x^*)$.
Arranging terms in equation (16):

$$\eta(1 - \frac{L\eta}{2}) \sum_{t=0}^{T-1} \mathbb{E}[\|\nabla F(x_t)\|^2] \leq \mathbb{E}[F(x_0)] - F(x^*) - \eta(1 - L\eta) \sum_{t=0}^{T-1} \mathbb{E}[\Delta_t] + 2L\eta^2\beta^2 T + L\frac{\eta^2}{2}\sigma^2 T. \quad (17)$$

Dividing equation (17) by $\eta(1 - L\frac{\eta}{2})T$:

$$\frac{1}{T} \sum_{t=0}^{T-1} \mathbb{E}[\|\nabla F(x_t)\|^2] \leq \frac{\mathbb{E}[F(x_0)] - F(X^*)}{\eta(1 - L\frac{\eta}{2})T} - \frac{(1 - L\eta)}{(1 - L\frac{\eta}{2})T} \sum_{T=0}^{T-1} \mathbb{E}[\Delta_t] + \frac{2L\eta\beta^2}{(1 - L\frac{\eta}{2})} + \frac{L\eta\sigma^2}{2(1 - L\frac{\eta}{2})} \quad (18)$$

Reconstruction errors are unbiased with respect to the direction of the gradient. Therefore, $\mathbb{E}[\Delta_t] = 0$.

$$\frac{1}{T} \sum_{t=0}^{T-1} \mathbb{E}[\|\nabla F(x_t)\|^2] \leq \frac{\mathbb{E}[F(x_0)] - F(X^*)}{\eta(1 - L\frac{\eta}{2})T} + \frac{2L\eta\beta^2}{(1 - L\frac{\eta}{2})} + \frac{L\eta\sigma^2}{2(1 - L\frac{\eta}{2})} \quad (19)$$

$$\frac{1}{T} \sum_{t=0}^{T-1} \mathbb{E}[\|\nabla F(x_t)\|^2] \leq \frac{\mathbb{E}[F(x_0)] - F(X^*)}{\eta(1 - L\frac{\eta}{2})T} + \frac{L\eta}{1 - L\frac{\eta}{2}}[2\beta^2 + \frac{\sigma^2}{2}] \quad (20)$$

Equation (20) shows the precise bound for DiffusedSplitFed-V1 during FP. Here, the first term is the optimization progress term and the second term is the error floor from bias and noise.
To convert it to the asymptotic form, we will take these two terms separately.

Term 1: Optimization progress: $\frac{\mathbb{E}[F(x_0)] - F(X^*)}{\eta(1 - L\frac{\eta}{2})T}$
This term captures the progress made in the $T$ communication rounds. It decays with $1/T$, and thus this part shrinks as the number of communication rounds increases. It depends inversely on $\eta$, and a slow convergence is expected when $\eta$ is small. In DiffusedSplitFed, $\eta$ is constant. So, the first term decays as $1/T$, and the asymptotic form is: $O(1/T)$.

Term 2: error floor from bias and noise: This term captures the lower bound on how small the gradient norm can get due to $\beta^2$, $\alpha^2$ and $\eta$. This term is proportional to $\beta^2$, $\alpha^2$ and $\eta$. Therefore, we can write it as: $O(\eta(\sigma^2 + \beta^2))$. And since $\beta^2 \sim \delta_t^{(1)} + \delta_t^{(2)}$, we can write: $O(\eta(\sigma^2 + \delta_t^{(1)} + \delta_t^{(2)}))$.

Therefore, the asymptotic form of the bound is:

$$\frac{1}{T} \sum_{t=0}^{T-1} \mathbb{E}[\|\nabla F(x_t)\|^2] \leq O(\frac{1}{T}) + O(\eta(\sigma^2 + \delta_d^{(1)} + \delta_d^{(2)})). \quad (21)$$

$$\frac{1}{T} \sum_{t=0}^{T-1} \mathbb{E}\|\nabla L(w_t)\|^2 \leq O(\frac{1}{T}) + O(\eta(\sigma^2 + \sum_{j=1}^{2} \delta_d^{(j)})) \quad (22)$$

The convergence bound for DiffusedSplitFed-V1 during FP is proved.
As $T$ grows, $O(\frac{1}{T})$ the term vanishes, and the dominant error terms are $\sigma^2$ and the variance term. Thus, we can say that the DiffusedSplitFed-V1 term converges to an error neighbourhood on the order of $\beta^2$ and $\sigma^2$ around the optimum. In addition, smaller reconstruction errors will lead to tighter convergence or to getting closer to the true optimum.

Similarly, in DiffusedSplitFed-V1, during FBP, we have four denoising steps, with LDDMs during the split learning process. In addition to the FP process, the third and fourth LDDMs apply during gradient denoising. Following the same steps as above, the precise bound for DiffusedSplitFed-V1 during FBP is:

$$\frac{1}{T} \sum_{t=0}^{T-1} \mathbb{E}[\|\nabla F(x_t)\|^2] \leq \frac{\mathbb{E}[F(x_0)] - F(X^*)}{\eta(1 - L\frac{\eta}{2})T} + \frac{L\eta}{1 - L\frac{\eta}{2}}[2\beta^2 + \frac{\sigma^2}{2}] \quad (23)$$

And the asymptotic form of the bound is:

$$\frac{1}{T}\sum_{t=0}^{T-1}\mathbb{E}[\|\nabla F(x_t)\|^2] \leq O(\frac{1}{T}) + O(\eta(\sigma^2 + \delta_d^{(1)} + \delta_d^{(2)} + \delta_d^{(3)} + \delta_d^{(4)})). \tag{24}$$

$$\frac{1}{T}\sum_{t=0}^{T-1}\mathbb{E}\|\nabla L(w_t)\|^2 \leq O(\frac{1}{T}) + O(\eta(\sigma^2 + \sum_{j=1}^{4}\delta_d^{(j)})) \tag{25}$$

The convergence bound for DiffusedSplitFed-V1 during FBP is proved.

### C.5.2 Convergence bound for DiffusedSplitFed-V2

In DiffusedSplitFed-V2, we have two denoising steps (in FP), with LDDMs during the split learning process. First, the features from FE are forward-diffused and then denoised on the server side by the first LDDM before being combined with global features and fed into the server model. Then, the server model's output is forward-diffused and then denoised on the BE, and this denoised output is combined with global features before inputting them into the BE model. Analogous to the DiffusedSplitFed-V1, we still have two reconstruction error terms in gradient computation, but V2 operates on isolated feature streams rather than aggregated features in V1. We denote these two reconstruction errors as follows:
$\epsilon_t^{(1)}$ is the reconstruction error of the first LDDM.
$\epsilon_t^{(2)}$ is the reconstruction error of the second LDDM.
Following the same steps as in DiffusedSplitFed-V1, we can prove that it reaches the following precise bounds: during FP and FBP (26) and asymptotic bounds: during FP (27) and during FBP (28), respectively.

$$\frac{1}{T}\sum_{t=0}^{T-1}\mathbb{E}[\|\nabla F(x_t)\|^2] \leq \frac{\mathbb{E}[F(x_0)] - F(X^*)}{\eta(1 - L\frac{\eta}{2})T} + \frac{L\eta}{1 - L\frac{\eta}{2}}[2\beta^2 + \frac{\sigma^2}{2}] \tag{26}$$

$$\frac{1}{T}\sum_{t=0}^{T-1}\mathbb{E}\|\nabla L(w_t)\|^2 \leq O(\frac{1}{T}) + O(\eta(\sigma^2 + \sum_{j=1}^{2}\epsilon_d^{(j)})) \tag{27}$$

$$\frac{1}{T}\sum_{t=0}^{T-1}\mathbb{E}\|\nabla L(w_t)\|^2 \leq O(\frac{1}{T}) + O(\eta(\sigma^2 + \sum_{j=1}^{4}\epsilon_d^{(j)})) \tag{28}$$

The convergence bounds for DiffusedSplitFed-V2 during FP and FBP are proved.

### C.5.3 Convergence bound for DiffusedSplitFed-V3

In DiffusedSplitFed-V3 also, we have two denoising steps (in FP), with LDDMs during the split learning process. First, the features from FE are forward-diffused and then denoised on the server side by the first LDDM and fed into the server model. Then, the server model's output is forward-diffused and then denoised on the BE. Analogous to the DiffusedSplitFed-V1 and V2, we still have two reconstruction error terms in gradient computation, but V3 operates without any external feature fusion, neither any ViT features nor global features. We denote these two reconstruction errors as follows:
$\zeta_t^{(1)}$ is the reconstruction error of the first LDDM.
$\zeta_t^{(2)}$ is the reconstruction error of the second LDDM.
Following the same steps as in DiffusedSplitFed-V1, we can prove that it reaches the following precise bounds: during FP and FBP (29) and asymptotic bounds: during FP (30) and during FBP (31), respectively.

$$\frac{1}{T}\sum_{t=0}^{T-1}\mathbb{E}[\|\nabla F(x_t)\|^2] \leq \frac{\mathbb{E}[F(x_0)] - F(X^*)}{\eta(1 - L\frac{\eta}{2})T} + \frac{L\eta}{1 - L\frac{\eta}{2}}[2\beta^2 + \frac{\sigma^2}{2}] \tag{29}$$

$$\frac{1}{T}\sum_{t=0}^{T-1}\mathbb{E}\|\nabla L(w_t)\|^2 \leq O(\frac{1}{T}) + O(\eta(\sigma^2 + \sum_{j=1}^{2}\zeta_d^{(j)})) \tag{30}$$

$$\frac{1}{T} \sum_{t=0}^{T-1} \mathbb{E}\|\nabla L(w_t)\|^2 \leq O(\frac{1}{T}) + O(\eta(\sigma^2 + \sum_{j=1}^{4} \zeta_d^{(j)})) \tag{31}$$

The convergence bounds for DiffusedSplitFed-V3 during FP and FBP are proved.

### C.5.4 Convergence bound for Perfect denoising/ Baseline SplitFed

During perfect denoising/ Baseline SplitFed all the reconstruction errors vanish. Therefore, substituting zero for the bias/ reconstruction loss terms in equations (20), (26), and (29), the precise bound and the asymptotic bound can be expressed as in equations (32) and (33) respectively.

$$\frac{1}{T} \sum_{t=0}^{T-1} \mathbb{E}[\|\nabla F(x_t)\|^2] \leq \frac{\mathbb{E}[F(x_0)] - F(X^*)}{\eta(1 - L\frac{\eta}{2})T} + \frac{L\eta\sigma^2}{2(1 - L\frac{\eta}{2})} \tag{32}$$

$$\frac{1}{T} \sum_{t=0}^{T-1} \mathbb{E}[\|\nabla F(x_t)\|^2] \leq O(\frac{1}{T} + \eta\sigma^2) \tag{33}$$

The convergence bound for perfect denoising/ Baseline SplitFed is proved.

### C.6 Conclusion-Convergence analysis

All DiffusedSplitFed variants converge under standard assumptions with additional bias from LDDM reconstructions. Improving reconstruction (reducing reconstruction errors) leads to tighter convergence.

## D  Ablation studies

Table 9 presents a comparative evaluation of the two feature fusion strategies we use when applied to DiffusedSplitFed-V1: attention-based and weighted fusion—under both FP and FBP configurations. The IoUs demonstrate that both feature fusion strategies contribute to effective feature integration; notably, the weighted feature fusion slightly outperforms attention-based fusion. This indicates its potential to enhance the segmentation performance.

Table 9: IoU comparison of feature fusion strategies in DiffusedSplitFed-V1: Attention-based vs. Weighted fusion under FP and FBP settings. **FP:** Latent diffusion only in the forward pass, **FBP:** Latent diffusion in both the forward & backward pass.

| Dataset | Attention-based feature fusion | | Weighted feature fusion | |
|---|---|---|---|---|
| | FP | FBP | FP | FBP |
| HAM10K | 0.893 | 0.908 | 0.897 | 0.912 |
| Blastocysts | 0.848 | 0.857 | 0.851 | 0.845 |
| FHPsAOPMSB | 0.885 | 0.858 | 0.887 | 0.860 |

