# OpenReview forum: "DiffusedSplitFed: Latent Diffusion and global feature fusion meet Split Federated medical image segmentation"
_TMLR — Withdrawn by Authors_

### Review · Reviewer_sUha · 2025-11-10

**Summary Of Contributions:**

This paper proposes a Split Federated Learning (SplitFed) framework for privacy-preserving medical image segmentation that integrates latent denoising diffusion models (LDDMs) at the client–server split points. The key idea is to inject and then denoise noise in latent feature space so that the exchanged representations are obfuscated for potential attackers but still informative for downstream segmentation. The authors design three architectural variants (V1–V3) that combine diffusion, denoising and feature perturbation. Extensive experiments on three medical-imaging segmentation datasets show that the proposed variants achieve IoU comparable to or better than a baseline SplitFed UNet. The paper further evaluates privacy via reconstruction attacks on intermediate features using UNet and CNN-Transformer attackers, and they show reduced reconstruction quality relative to the baseline SplitFed. Finally, the authors sketch a convergence analysis, adapting standard FL/SL proof techniques to the case where gradients are computed on denoised latent features.

**Strengths.**
1. Well-motivated attempt to combine SplitFed and latent diffusion for privacy-preserving segmentation, an area that is currently underexplored.
2. Solid empirical evaluation on multiple datasets, with comparisons to both a baseline SplitFed pipeline and diffusion-based medical segmentation methods; results are generally favorable.

**Weaknesses.**
1. The methodology section is quite complex, and more importantly, some design choices (e.g., choice of fusion strategy, diffusion hyper-parameters, why these three specific variants) seem to be very heuristic and would benefit from clearer justification and ablation.
2. The paper could better articulate the incremental novelty relative to recent federated diffusion and SplitFed-diffusion works (e.g., FedDiffuse/CollaFuse) beyond "first with LDDM for SplitFed medical segmentation".

**Audience:**

Yes

**Audience Explanation:**

This paper sits at the intersection of topics including federated learning, privacy-preserving ML, and medical imaging, by combining SplitFed architectures with latent diffusion for privacy-aware segmentation, and by empirically exploring privacy–utility–complexity trade-offs on real medical datasets.

Even though some aspects are incremental (e.g., using LDDMs at split points rather than entirely new diffusion formulations), the work provides a concrete and reproducible framework that practitioners in medical FL and SplitFed could build upon. The empirical findings regarding how different diffusion placements and global feature fusion strategies affect both IoU and reconstruction leakage should be of practical interest to this community.

**Broader Impact Concerns:**

The work targets privacy-preserving medical image segmentation in federated/split settings. On the positive side, better protection of intermediate representations could meaningfully reduce the risk of patient data leakage when institutions collaborate on training segmentation models, which is highly beneficial for real-world deployment in healthcare.

However, the approach does not provide formal privacy guarantees, and an over-reliance on empirical reconstruction metrics might lead practitioners to overestimate the level of protection. The authors should explicitly caution that more powerful or different attackers may still extract sensitive information, and that DiffusedSplitFed should ideally be combined with regulatory and system-level safeguards.

**Claims And Evidence:**

Yes

**Claims Explanation:**

Overall, the core claims, namely DiffusedSplitFed variants can match or improve segmentation performance over a baseline SplitFed UNet and diffusion-based baselines under the evaluated settings, are reasonably supported by the experiments. The datasets, metrics (IoU), and baselines are described, and the tables clearly show that several variants (especially V3) outperform the baseline SplitFed across datasets, while also being competitive with or better than GMS and LDSeg in both centralized and SplitFed configurations.

The privacy claims are supported by reconstruction-attack evaluations. Specifically, the quantitative metrics (PSNR, SSIM, FID, MSE) consistently indicate that reconstructed images from DiffusedSplitFed features are of lower quality than those from baseline SplitFed, suggesting reduced leakage. However, these are necessarily conditional on the chosen attackers and do not constitute formal guarantees.

**Requested Changes:**

1. Rephrase claims such as "ensuring privacy" or "privacy guarantees" to make clear that the paper provides empirical evidence of reduced reconstruction quality under specific attackers, rather than formal differential-privacy-style guarantees.

2. Where possible, compare to additional threat models or attacks (e.g., membership inference) or at least discuss why the chosen attackers are representative and what kinds of leakage may still remain.

3. Provide more ablations on key hyper-parameters of the diffusion process (e.g., noise level, number of diffusion steps, weight of reconstruction losses) and show how they affect both IoU and reconstruction metrics.

4. For V1–V3, disentangle the contributions of global feature fusion, dual conditioning, and latent diffusion more systematically; currently, some of these are intertwined.

5. Expand the discussion on how DiffusedSplitFed differs from and improves upon existing federated diffusion frameworks such as FedDiffuse, CollaFuse, Phoenix, etc., beyond the claim of being first in SplitFed medical segmentation. It would be useful to frame the contribution as a systematic design/engineering study in this space.

---

### Review · Reviewer_EDgw · 2025-11-12

**Summary Of Contributions:**

**(C1) Proposal of DiffusedSplitFed (Sec. 2)**

DiffusedSplitFed integrates LDDMs at both the forward and backward split points of SplitFed to obfuscate transmitted latent representations and enhance privacy. The design combines dual conditioning and global feature fusion, and includes three variants: V1 (dual + global), V2 (global only), and V3 (lightweight version).

**(C2) Comprehensive empirical evaluation of the privacy–utility trade-off (Sec. 3)**

Across three medical-image datasets (HAM10K, Blastocysts, and FHPsAOPMSB), DiffusedSplitFed performed well compared with baselines in segmentation performance while strengthening robustness to reconstruction attacks. Notably, FBP maintains segmentation performance (IoU) comparable to FP while further improving privacy. V2/V3 are strongest on privacy, whereas V1 is effective when accuracy is prioritized.

**(C3) Convergence analysis of DiffusedSplitFed (Sec. 5)**

Under certain assumptions, a convergence analysis is provided for DiffusedSplitFed (V1/V2/V3).

**Audience:**

Yes

**Audience Explanation:**

In this field, the standard privacy-preserving approach is to incorporate Differential Privacy into Federated Learning (FL-DP). This work pursues an alternative route, which may attract interest from at least some individuals in TMLR's audience.

**Broader Impact Concerns:**

While the paper empirically demonstrates robustness to reconstruction attacks from intermediate representations, the attack coverage is narrow, and the risk of privacy leakage is not rigorously shown.

**Claims And Evidence:**

No

**Claims Explanation:**

**(1) Limited empirical evidence**

**Missing broader benchmarks.** While the primary application is medical image segmentation, the paper provides no formal privacy guarantee and therefore requires broader empirical validation on additional benchmarks (e.g., image classification, generation, and NLP tasks).

**Limited attack coverage.** The evaluation centers on reconstruction attacks only and does not assess other critical vectors such as model inversion, gradient leakage, or membership inference.

**Non-IID and scalability are under-explored.** Robustness to client heterogeneity (non-IID data) and scaling with the number of clients appears untested.

**Insufficient hyperparameter tuning.** Key hyperparameters are underexplored, including noise schedulers and Adam learning rates.
No error-bar reporting. Results lack error bars across multiple random seeds.

**(2) Missing theoretical evidence in privacy-preserving**

Empirical evidence limited to reconstruction attacks on limited domain benchmark tests may be insufficient; it is preferable to provide theoretical evidence for the privacy-preserving effectiveness of DiffusedSplitFed.

**Requested Changes:**

**(1) Motivation for the proposed method.**

Differential Privacy for Federated Learning (FL-DP) is the standard privacy-preserving approach in this area. Could you elaborate on your motivation for instead integrating diffusion models to achieve privacy-preserving? In addition, in the Introduction, the statement “they may fail to preserve semantic fidelity in high-dimensional tasks such as medical image segmentation” is unclear—please explain furthermore. Likewise, the claim that FL-DP methods “often introduce significant computational overhead” needs clarification: what is indicated?

**(2) Lacks in experiments**

As I noted above, the empirical evidence is limited by a) narrow benchmarks, (b) restricted attack coverage, (c) non-IID and NW scalability are unexplored, (d) hyperparameter tuning, and (e) error-bar reporting.

---

### Review · Reviewer_DwQv · 2025-11-15

**Summary Of Contributions:**

Paper Summary：

This paper proposes the DiffusedSplitFed framework, introducing LDDM (Digital Feature Derivatives) into federated segmentation tasks for the first time, specifically for privacy-preserving medical image segmentation. This method incorporates LDDM into the forward and backward propagation between the client and server to perturb and denoise intermediate features during transmission. Three architectural variants, v1-v3, are designed to explore biconditional global feature fusion, global feature fusion, and a lightweight diffusion mechanism, respectively, and their performance is validated on corresponding medical image segmentation datasets. This innovative approach provides a fresh perspective for medical image segmentation tasks.

Strengths：
(1) This paper proposes combining LDDM with the SplitFed framework. By leveraging the inherent learnable "noise-denoising" process of the diffusion model, it can both destroy the semantic information of features to protect privacy and retain sufficient information for downstream segmentation tasks through the learnable denoising process.
(2) The article proposes three schemes for DiffusedSplitFed, which provide a flexible trade-off between privacy, performance and complexity. This modular design enables the framework to adapt to different real-world application scenarios and needs.
(3) The experimental section conducted comprehensive experimental verification, including segmentation performance, countermeasures evaluation tests, and efficiency analysis, to validate its effectiveness and superiority.

Weakness：
(1) Although the authors provided a framework diagram in Fig. 2, we cannot determine the origin of the Global Feature from either the diagram or the algorithm formula.
(2) The comparison method is too limited. The paper mainly compares with the basic SplitFed methods (GMS, LDSeg), but does not compare with mainstream privacy protection technologies under the SplitFed framework.
(3) Regarding Table 4, "Comparison of segmentation performance," why can't the GMS and LDSeg Baseline SplitFed methods incorporate the LDDM structure, and why are the experimental results for the corresponding rows unavailable?
(4) The paper describes the various variants of DiffusedSplitFed in great detail in Section 2. However, the corresponding algorithms (Algorithms 1-3) and framework diagrams (Figures 1-3) are placed in later chapters, which makes it inconvenient for readers to understand the details of the methods.

**Audience:**

Yes

**Audience Explanation:**

See above

**Claims And Evidence:**

Yes

**Claims Explanation:**

See above

**Requested Changes:**

See above

---

### Note · Authors · 2025-11-17

**Comment:**

Dear reviewers and the Action editor,

We kindly request to withdraw this submission from TMLR.
We sincerely appreciate the Action Editor’s and reviewers’ valuable insights, comments, and suggestions. Based on their feedback, we plan to undertake a major revision of the manuscript.

Authors of paper 6011.

**Withdrawal Confirmation:**

I have read and agree with the venue's withdrawal policy on behalf of myself and my co-authors.